# Safety of Fibrinogen Concentrate for Correcting Perioperative Bleeding-Associated Hypofibrinogenemia in Adults: A Single-Center Experience

**DOI:** 10.3390/jcm13196018

**Published:** 2024-10-09

**Authors:** Manuela Gomes, Miguel Ângelo-Dias, Jorge Lima

**Affiliations:** 1Transfusion Medicine Department, Hemovida, Hospital da Luz Lisboa, Luz Saúde, 1500-650 Lisboa, Portugal; amdrgv@gmail.com; 2NOVA Medical School, Faculdade de Ciências Médicas, NMS, FCM, Universidade Nova de Lisboa, 1169-056 Lisboa, Portugal; miguel.dias@nms.unl.pt; 3CHRC, NOVA Medical School, Faculdade de Ciências Médicas, NMS, FCM, Universidade Nova de Lisboa, 1169-056 Lisboa, Portugal; 4Department of Obstetrics and Gynecology, Hospital da Luz Lisboa, Luz Saúde, Avenida Lusíada 100, 1500-650 Lisboa, Portugal

**Keywords:** fibrinogen concentrate, perioperative care, hypofibrinogenemia, thrombosis, haemostasis

## Abstract

**Background:** Surgery often leads to bleeding associated with hypofibrinogenemia. Supplementation with fibrinogen concentrate appears to be effective and safe, although findings from studies are inconsistent. The primary aim of this study was to assess the safety of fibrinogen concentrate during the perioperative period. **Methods:** This single-centre, prospective, observational study included adult patients undergoing scheduled or emergency surgery related to bleeding coagulopathy and the administration of fibrinogen concentrate. Patients were followed until their discharge from the institution. Comprehensive data were collected, including age, sex, type of surgery, associated comorbidities, anticoagulant and/or anti-aggregating therapy, and the number of blood transfusions. Laboratory data on plasma fibrinogen concentration, haemoglobin, and platelet count before and after surgery were also collected. The primary outcomes were the mortality rate at discharge and any reported thrombotic or thromboembolic events, including deep vein thrombosis, pulmonary embolism, and myocardial infarction. **Results:** The study included 91 adult patients who had undergone surgery, with 29 surgeries (32%) conducted in an emergency setting. The mean age was 59.2 years, and 53.8% were male. Major bleeding occurred in 29 cases, mainly in older males and those on anticoagulant therapy. The pre-operative fibrinogen level averaged 161 mg/dL, and the average dosage of fibrinogen concentrate administered was 2.7 g. Eight patients died (8.8%), mostly due to septic or cardiogenic shock, with deaths being more frequent in emergency settings. Thromboembolic events occurred in eight patients, none of whom died. No additional adverse events directly related to the administration of fibrinogen concentrate were reported. **Conclusions:** Our findings suggest a favourable safety profile for fibrinogen concentrate in surgical patients, as evidenced by a low incidence of deaths and thromboembolic events, which were primarily attributed to other factors. Future research should strive to increase statistical robustness to further illuminate clinically significant patient safety measures.

## 1. Introduction

Surgery often results in severe bleeding, necessitating precise and effective haemostatic management. Central to this process is fibrinogen, the main plasma protein coagulation factor acknowledged as an early participant in events leading to clot formation. Primarily recognized for its role as the fibrin precursor, it becomes cleaved by thrombin, and fibrinogen-derived fibrin monomers then polymerize to form a net that entraps cells, including platelets, establishing the clot’s foundation. Additionally, fibrinogen significantly contributes to platelet aggregation by binding to GP IIb/IIIa receptors on activated platelets. Furthermore, as an acute-phase reactant, fibrinogen modulates inflammatory cellular responses, underscoring its crucial role in maintaining primary and secondary haemostasis [1,2].

Synthesized in hepatocytes, fibrinogen’s plasma concentration typically ranges from 1.5 to 4.5 g/L. However, in circumstances such as pregnancy, these levels can become even higher [1,3]. Various factors influence normal fibrinogen levels, including physiological (age, sex), pathological (hepatic disease), and lifestyle factors (smoking) [1]. Conversely, bleeding episodes can result in fibrinogen loss, dilution, and consumption, compounded further by factors such as hypothermia and acidosis, which can impair its function [3,4]. Therefore, patients experiencing significant bleeding during surgery are at an increased risk of developing hypofibrinogenemia, where fibrinogen levels drop to critically low levels.

The best approach to fibrinogen supplementation remains under debate, with three main methods: plasma, cryoprecipitate, and fibrinogen concentrate.

Plasma is not the optimal source for fibrinogen replacement. There are various types of plasma, namely solvent detergent plasma and fresh frozen plasma (FFP), with fibrinogen levels typically between 2 and 2.9 g/L. Some FFP units may have levels as low as 1 g/L. As a result, substantial volumes are necessary to achieve a substantial correction in fibrinogen levels. Additionally, plasma needs to be ABO-compatible and thawed, necessitating preparation time and equipment [5,6]. Cryoprecipitate, while containing higher fibrinogen levels than FFP, has an uncertain fibrinogen concentration and also contains other coagulation factors. It is usually administered as a pooled product originating from 6 to 10 units of blood, increasing the recipient’s exposure to multiple donors. The primary concern with cryoprecipitate is its safety profile since it does not undergo antiviral processing and is not available for use in several European countries due to safety concerns [4].

Fibrinogen concentrate is a rigorously purified, pathogen-inactivated, and lyophilized product derived from plasma. This results in a sterile, stable product that provides a pre-determined and standardized amount of fibrinogen replacement per vial. Given these qualities, fibrinogen concentrate has become a go-to intervention for correcting hypofibrinogenemia-associated coagulopathy during the perioperative period. It allows for accurate and quick restoration of fibrinogen levels, thus promoting successful clot formation and platelet function. Pharmacovigilance data accumulated over the decades suggest that fibrinogen concentrate supplementation, in cases of congenital and acquired fibrinogen deficiency, does not lead to adverse events [7,8]. However, the trigger and target levels for fibrinogen supplementation vary and appear to depend on multiple factors. That said, several guidelines propose that fibrinogen levels between 1.5 and 2.0 g/L should be adjusted via supplementation [9,10].

Although numerous studies have reviewed the use of fibrinogen concentrate in correcting hypofibrinogenemia and improving haemostasis [7,11,12,13,14], further investigation is needed to thoroughly assess its efficiency and safety profile. A significant concern is that fibrinogen concentrate, a procoagulant, is administered to patients already posing risk factors for thrombotic or thromboembolic events. While most clinical trials suggest fibrinogen concentrate’s safety and effectiveness, inconsistencies in study designs, small sample sizes, varying institutional protocols, pre-existing conditions and medicines in patients, imprecise triggers and targets for fibrinogen supplementation, and a variation in dosages, often unrepeated in complicated and major surgeries, contribute to inconsistent findings. Moreover, the clinical and lab-based decisions on when and how to supplement fibrinogen vary widely. The conflicting data in the existing literature emphasize the need for a rigorous evaluation of the efficiency and safety of administering fibrinogen concentrate in the perioperative setting [13,14,15,16,17,18,19,20,21,22,23].

The primary objective of this study was to assess the safety of fibrinogen concentrate when administered within therapeutic ranges during the perioperative period to correct bleeding-associated hypofibrinogenemia in adults. Conducting a real-world analysis, this research aimed to provide valuable insights into the practical use and safety profile of fibrinogen concentrate, addressing a vital aspect of managing haemostasis in surgical and bleeding scenarios.

## 2. Materials and Methods

### 2.1. Study Design

This was a prospective observational study conducted at Hospital da Luz Lisboa, Portugal, between July 2022 and December 2023. The study included all adult patients (≥18 years of age) who underwent scheduled or emergency surgery and required the administration of fibrinogen concentrate for correcting perioperative bleeding-associated hypofibrinogenemia. These patients were followed until their discharge from the institution. Fibrinogen concentrate doses (Haemocomplettan CSL Behring, Marburg, Germany) within the range of 25–50 mg/kg/dose were administered to all the patients in this study.

Patients with hereditary bleeding disorders that lead to fibrinogen deficiency or abnormal function were excluded. Patients exhibiting bleeding conditions accompanied by acquired hypofibrinogenemia associated with postpartum trauma, or any other medical condition, such as gastrointestinal bleeding, were also excluded.

### 2.2. Bleeding Protocol

The approach to managing bleeding patients adhered to our institution’s protocol, grounded on the guidelines for major haemorrhages issued by the Portuguese Directorate of General Health [24]. This protocol underlines the importance of multidisciplinary communication and the critical initial steps in patient care. The essential initial steps comprise the following: the swift identification of surgical causes of bleeding and the implementation of necessary control measures; the reappraisal of the patient’s therapeutic and medical history, particularly in emergency cases; the determination of haemoglobin and platelet levels; the assessment of coagulation status via standard coagulation screens or through the use of point-of-care test results (ROTEM^®^ sigma—TEM Innovations, Munich, Germany); and the stabilization of the patient’s basic physiological conditions, including addressing hypothermia, acidosis, hypocalcaemia, and anaemia.

This protocol outlines a sequential approach for addressing specific aspects of coagulopathy. Initially, if a clinical or laboratory suspicion of hyperfibrinolysis arises, correction with tranexamic acid is initiated at a dose of 20–25 mg/kg, followed by the administration of fibrinogen if a deficiency is observed, using fibrinogen concentrate (Haemocomplettan CSL Behring, 25–50 mg/kg). Thrombocytopenia is corrected with platelet concentrate, while a deficit of thrombin or other coagulation factors is addressed with prothrombin complex concentrate (PCC) (Octaplex Octapharma, 20–30 UI/kg) or FFP at a dose of 12–15 mL/kg. Treatments with haemostatic agents can be repeated as needed to control coagulopathy, always guided by standard coagulation tests and/or point-of-care test results and clinical evaluation.

Major bleeding was defined according to the volume of blood loss (one total blood volume (TBV)/24 h, or >50% of the TBV/3 h), ongoing bleeding rhythm (150 mL/min), and the number of transfused blood units.

The cut-off values considered for each transfused agent were as follows: RBC if <7 g/dL or 8 if heart disease; platelets if <50 × 10^9^/L and persistent active bleeding; FC if <200 mg/dL or blood loss > 1–1.5 L and ongoing bleeding, or according to ROTEM results; and PCC if international normalized ratio (INR) or activated partial thromboplastin time (aPTT) > 1.5 and acute active bleeding, or volume blood loss 150–200%, or 100% and liver disease. Repeated doses were applied depending on bleeding severity and the results of coagulation assessments after each treatment cycle.

Patients with anticoagulant or antiplatelet therapy were managed depending on the type of antithrombotic therapy, the time until surgery, and the risk of thromboembolic events. Namely, this occurred when surgery could not be delayed, and according to laboratory results, the procedure was as follows: for vitamin K antagonists, vitamin K and/or PCC (15–30 IU/kg) were administered. For rapid reversal of direct oral anticoagulants, PCC was used at 25–50 IU/kg. In this group of patients, we had no need to use idarucizumab for dabigatran reversal. For antiplatelet therapy, a pool platelet concentrate was considered according to the clinical situation. Clinical indications for both therapies were formal indications, such as atrial fibrillation and coronary disease, among others.

The protocol was approved by the Institutional Ethics Committee, and informed consent was obtained from each study participant.

### 2.3. Data Collection

Data collection and patient follow-up began at the anaesthesiology appointment and concluded upon discharge from the hospital. Comprehensive data were gathered from each patient, including age, sex, type of surgery, associated comorbidities, anticoagulant and/or antiplatelet therapy, the number of red blood cell, and platelet concentrate units administered, as well as the total doses of fibrinogen concentrate, prothrombin complex concentrate, antithrombin concentrate, and FFP administered.

Additionally, when available, data regarding plasma fibrinogen concentration (determined by the Clauss method and/or results from viscoelastic testing), last haemoglobin, and platelet count before surgery, as well as the lowest haemoglobin and platelet count in the post-operative period were also collected.

The mortality rate at discharge, as well as all reported thrombotic or thromboembolic events, including deep vein thrombosis (DVT), pulmonary embolism (PE), myocardial infarction, and any other reported or clinically suspected thromboembolic event, were also recorded and documented. PE and DVT were monitored with ultrasound scan, D- Dimer Test, ECG chest X-ray, and clinical signs and symptoms. Most patients continued to be followed for a variable amount of time depending on personal residence and clinical outcome, but most of them were monitored for more than 3 months.

In addition, the presence of additional risk factors for thrombotic/thromboembolic events at the baseline, such as obesity, smoking status, family and past personal history of VTE, the use of oral contraceptives, hormone replacement therapy, and glucocorticoids, were also noted. Given that these factors might serve as confounding variables, these data facilitated further statistical stratification analysis.

### 2.4. Outcomes

The primary outcomes of interest in this study were overall mortality and the incidence of thromboembolic events at discharge. These events included myocardial infarction, DVT, PE, or stroke, all of which could potentially be associated with the administration of fibrinogen concentrate.

The secondary outcomes focused on additional adverse events reported, such as anaphylaxis, hypersensitivity, or allergic reactions, during the study period. They also focused on the total number of allogeneic blood components administered until discharge and the overall length of hospital stay.

### 2.5. Statistical Analysis

Categorical variables are presented as absolute frequencies and percentages; their associations were analysed using Fisher’s exact test. Continuous data, if normally distributed, are expressed as the mean and standard deviation (SD); if not normally distributed, they are expressed as the median and interquartile range (IQR). The normality of these data was assessed by visually inspecting QQ plots of the residuals and by employing the D’Agostino–Pearson normality test when needed.

For data that are normally distributed, comparisons between two unpaired groups were conducted using the unpaired *t*-test or the unpaired *t*-test with Welch’s correction, as appropriate. If not, the non-parametric Mann–Whitney test was employed. Correlations were determined using Spearman’s correlation coefficient.

The test used in each case is specified in the respective figure and table legends. Any analysis with a *p*-value less than 0.05 was considered significant. Statistical analyses were conducted using GraphPad Prism v10.1.2 for Windows (GraphPad Software, Boston, MA, USA). Visualizations were created using Microsoft Power BI software v2.131.1203.0 for Windows (Microsoft Corporation, Redmond, WA, USA).

## 3. Results

### 3.1. Demographics, Comorbidities, and Concomitant Medication

A total of 91 adult patients were included in the study. Of these, 29 were urgent or emergency surgeries. Details of patient demographics and clinical characteristics are provided in Table 1.

The mean age of the recruited patients was 59.2 ± 18.2 years, with an average weight of 71.6 ± 15.3 kg and a mean body mass index (BMI) of 25.1. Of these, forty-nine (53.8%) were males. Twenty patients were active smokers, and sixty-five had at least one related comorbidity. Before surgery, 36 patients were given medication that could potentially impair haemostasis, including anticoagulant and/or antiplatelet therapies.

Eleven patients (12.1%) had a history of previous thromboembolic events, and only three had a family history of such events.

### 3.2. Surgery and Bleeding

The types of surgery conducted in the study were predominantly cardiovascular (37%) or abdominal (36%) (Figure 1). Among the remaining surgeries, ten were spinal surgeries, two were orthopaedic procedures, and twelve, labelled as ‘other’, included thoracic and plastic surgeries.

Abdominal surgeries include procedures related to gynaecological, gastric, and prostatic issues, while cardiac surgeries primarily treat patients with valvular or coronary heart disease. The median length of hospital stay was 6 days, with a range of 1 to 80 days.

During the perioperative period, 29 major bleeding cases were documented, either during the intervention or within 48 h post-surgery (Figure 2). Of this cohort, 72% were male and older (*p* = 0.014), while 35% undertook anticoagulant therapy, received more units of RBC (*p* < 0.001), and fibrinogen concentrate (*p* = 0.028). Subsequently, they spent more days in the hospital (*p* = 0.005). Within these 29 patients, 11 patients received a platelet pool concentrate, 9 patients received PCC, 27 patients received RBC transfusions, and 14 patients received FFP. Only 5 patients were treated with all the aforementioned haemostatic agents. Notably, nearly half of the patients subjected to emergent interventions experienced major bleeding (*p* = 0.030).

### 3.3. Perioperative Haemostasis and Blood Component Transfusions

The mean plasma fibrinogen levels, measured using the Clauss method, were 161 mg/dL at the decision point to administer fibrinogen concentrate, with five patients having levels below 100 mg/dL. Among patients who had point-of-care testing available at this time, seven had an FIBTEM MCF (Maximum Clot Firmness) below 9 mm, with the lowest value observed at 3 mm. The average dose of fibrinogen concentrate administered was 2.7 g, ranging from 1 to 7 g. All doses were calculated based on patient weight and fell within the recommended ranges, as previously described.

Before surgery, the mean haemoglobin level was measured at 12.9 ± 2.1 g/L, with the lowest post-operative level recorded at 7.9 ± 1.6 g/L. This represents an average decrease of 5 g/L. Similarly, the highest average pre-operative platelet level was 216 × 10^9^/L, which fell to post-operative levels of 139 × 10^9^/L, which is nearly a half. Following the hospital’s bleeding management protocol, 26 patients received a platelet pool concentrate, 13 were given prothrombin concentrates, and 36 were administered FFP. These are detailed in Table 1.

Seventy-one patients received RBC transfusions, with a median of two units administered per patient. Among these, 20 patients received more than four units of RBC. Nineteen patients did not require any transfusions with blood components. These included three cardiovascular, five abdominal, five neurosurgical, one orthopaedic, and five thoracic surgeries, as bleeding was effectively managed using fibrinogen concentrate.

### 3.4. Primary Outcomes

#### 3.4.1. Death

Eight patients from the study group succumbed following their surgeries, which translated to a mortality rate of 8.8%. The main causes of death were septic shock (n = 4), cardiogenic shock (n = 2), and hypovolemic shock (n = 2), all leading to multiple organ failure. The majority of deaths (seven of eight) occurred in patients who had undergone abdominal surgeries in contrast to only one resulting from cardiac surgery (*p* = 0.027). Of the patients who died, two received multiple transfusions, and none of the eight experienced any thromboembolic events. A significant association was found between the surgery setting and mortality (Figure 3), indicating a higher likelihood of deaths in emergent settings (20.7%) compared to non-emergent settings (3.2%). Furthermore, we noted that patients who passed away were older (*p* = 0.037), had lower fibrinogen levels at the time of the decision to administer fibrinogen concentrate (*p* = 0.018), and were recipients of multiple blood component transfusions. No significant variations were seen in the number of fibrinogen concentrate doses given between the two groups.

#### 3.4.2. Thromboembolic Events

Eight patients experienced at least one thromboembolic event post-surgery, with an instance of one patient enduring two such events (Figure 4). These occurrences included the following: one myocardial infarction, two deep vein thromboses, two pulmonary embolisms, and four strokes. Within this group, two patients also underwent multiple transfusions. Importantly, none of these patients died, and all were discharged from the hospital. Patients who experienced thromboembolic events received a greater quantity of RBC units (*p* = 0.043) and had lengthier median hospital stays before discharge (*p* = 0.046). In our research sample, half of the patients (n = 4) who manifested thromboembolic events had prior incidences of such events, in contrast to only 8.4% of the patients unaffected by such events (n = 7; *p* = 0.007). Only two patients experienced thromboembolic events and major bleeding. We observed no significant disparities in the fibrinogen concentrate doses administered across both groups.

### 3.5. Secondary Outcomes

In this group of 91 patients, no adverse events directly associated with the administration of fibrinogen concentrate, such as anaphylaxis, hypersensitivity, or allergic reactions, were reported.

Older patients (over 65 years) typically necessitated longer hospital stays, with a median duration of 10 days compared to 5 days required for younger patients (*p* < 0.001). Moreover, age is inversely correlated with pre-surgery platelet levels (r = −0.253; *p* = 0.026). Conversely, age is positively correlated with the amount of RBC units administered (r = 0.281; *p* = 0.007) and the length of hospital stay (r = 0.339; *p* = 0.002).

Similarly, patients with a personal history of previous thromboembolic events had lower post-surgery haemoglobin levels (*p* = 0.033) and required fewer units of platelets (*p* = 0.039) and FFP (*p* = 0.030).

Twenty-nine surgeries (32%) occurred in an emergency setting, of which 19 (66%) were abdominal procedures. Patients admitted under these circumstances exhibited significantly lower haemoglobin levels before (*p* = 0.003) and after surgery (*p* < 0.001). They also received more RBC units (*p* < 0.001), FFP (*p* = 0.011), fibrinogen concentrate (*p* = 0.019), and experienced extended hospital stays (*p* < 0.001).

We discovered a tendency toward a positive correlation between BMI and fibrinogen levels (r = 0.326; *p* = 0.053; Figure 5). Additionally, there was a negative correlation between lower post-operative haemoglobin levels and the number of RBC units (r = −0.569; *p* < 0.001), as well as the amount of fibrinogen concentrate administered (r = −0.356; *p* = 0.001), and the length of hospital stay (r = −0.386; *p* < 0.001).

## 4. Discussion

Fibrinogen concentrate is widely used in multiple countries to treat acquired hypofibrinogenemia in various clinical scenarios, including surgery [7]. Although many studies have validated its efficacy and safety, there are still conflicting data.

During significant blood loss, fibrinogen levels can decrease rapidly, a condition made worse by hyperfibrinolysis, severe shock, trauma, or acidosis. Moreover, compromised fibrinogen synthesis due to liver disease or hypothermia can contribute to hypofibrinogenemia [3,4].

This prospective observational study concentrated on the safety of administering fibrinogen concentrate in non-trauma and non-obstetric surgical patients since the inherent pathophysiology of coagulopathy could vary from standard surgery. This study primarily considered mortality and thromboembolic events, aiming to encapsulate a real-world approach to bleeding in relation to hypofibrinogenemia.

Despite decades of pharmacovigilance demonstrating a low risk of adverse events in both congenital and acquired fibrinogen deficiencies, concerns persist regarding the potential thromboembolic complications associated with fibrinogen concentrate administration. Adverse events linked to fibrinogen concentrate can include anaphylaxis, hypersensitivity, allergic reactions, thromboembolic complications, and potential virus transmission [7,8]. The latter could also stem from blood component administration or patient behaviour. These are rare occurrences that were not observed in our study.

Our study, which encompassed 91 surgical patients, did not reveal any noticeable associations between the administration of fibrinogen concentrate and mortality among the eight patients who died. Interestingly, six of these patients were admitted in an emergency context with significant comorbidities. Additionally, an association was observed among those who received multiple transfusions and underwent abdominal surgery, although causality remained unclear.

Similarly, eight patients experienced thromboembolic events after surgery, but they were all alive upon discharge from the hospital. Contrary to other reports, there was an equal distribution of sexes among these patients [7,8]. Confounding variables such as concurrent illnesses and the personal or familial history of thrombotic disease could provide alternate explanations for these thromboembolic events. Of these eight patients, only one had no associated comorbidities. Among the remaining seven, one had a positive family history of thrombotic disease. These patients also presented with diverse conditions, including prior thromboembolic events in two cases, a history of COVID-19 infection, cancer, and atrial fibrillation, among others. In this study, a few patients were recruited with conditions that may affect fibrinogen levels. In fact, only one patient had previously known non-congenital hypofibrinogenemia (~90 mg/dL). One patient had a liver transplant 25 years ago but presented with normal fibrinogen levels before surgery. Another patient had chronic liver disease and showed major bleeding after surgery. Regarding known prior haematological malignancies, one patient had myelodysplastic syndrome, one had Non-Hodgkin’s Lymphoma, and one had Chronic Lymphatic Leukemic, which also developed an episode of major bleeding after surgery.

Elevated plasma fibrinogen levels have been associated with thrombotic states across various clinical scenarios, such as venous thromboembolism, the rupture of arterial atherosclerotic plaque, and intracardiac thromboembolism [1]. Administering fibrinogen concentrate to achieve haemostasis in patients with major bleeding could disrupt the haemostatic equilibrium and potentially induce a thrombotic state due to its ability to increase plasma fibrinogen levels.

Research from animal and patient trauma models indicates that fibrinogen concentrate may temporarily heighten endogenous thrombin activity at the wound site, stemming from an elevated fibrinogen concentration in the plasma. However, this effect on plasma fibrinogen levels is fleeting, with no observed suppression of endogenous fibrinogen synthesis [25,26].

The transient effect of fibrinogen concentrate could be due to fibrinogen consumption or metabolism. For example, a study involving patients undergoing coronary artery bypass graft surgery showed an increase in D-dimers about 2 h after the administration of fibrinogen concentrate, supporting this idea [27].

Similarly, another study conducted on trauma patients found that plasma fibrinogen levels on day 3, following fibrinogen concentrate administration, resembled those observed in the control group. This observation aligns with the acute-phase response nature of fibrinogen. Its levels fluctuate in response to physiological stressors, reflecting a return to baseline levels over time [27].

The primary objective of our study was not to assess the efficacy of fibrinogen concentrate, which depends on various factors. These include the clinical context, institutional protocols, control of crucial steps, such as the early recognition of significant bleeding, effective team communication, and accepted targets for controlling bleeding coagulopathy, among others [6,15,28].

In terms of safety, which was our study’s primary focus, our findings suggest no association between the administration of fibrinogen concentrate in the perioperative setting and an increased risk of thromboembolic events. Of the eight patients who experienced thromboembolic events, each case had other strong factors contributing to these clinical occurrences. This suggests that fibrinogen concentrate administration cannot be solely attributed to these events.

Secondary outcomes align with some conclusions drawn by other authors [29,30,31]. Age appears to be a risk factor for the necessity of more units of RBC and extended hospital stays. Moreover, patients who endured thromboembolic events during the follow-up phase also had longer hospital stays and received more units of RBC. Our findings indicate that age, the incidence of thromboembolic events, and emergency surgeries are connected to a higher need for blood transfusions and longer hospital stays. Even though the use of fibrinogen concentrate did not seem to heighten the risk of thromboembolic events, patients requiring higher doses of fibrinogen concentrate also received more units of RBC. This finding aligns with a systematic review analysing the use of fibrinogen concentrate for haemorrhages in emergencies [32]. We would also like to underscore the positive correlation between BMI and fibrinogen levels at the deciding point.

We acknowledge several limitations to our study, including the small number of patients recruited, the variety of surgical procedures with differing underlying pathophysiological mechanisms for haemorrhage, and possible confounding variables like emergent situations and comorbidities. We also chose to confine our study protocol to hospital discharge, given the temporary effect of fibrinogen concentrate administration and the potential emergence of other confounding variables during an extended follow-up period in such a varied population.

Nevertheless, our study provides valuable insights into the management of bleeding complications in real-world settings outside the controlled environment of a randomized controlled trial. It underscores the need for special clinical caution in older patients, those undergoing emergency surgeries, the recipients of multiple blood transfusions, and patients with a history of thromboembolic events. Although there were insufficient data for statistical analysis, patients with critically low fibrinogen levels at the time of the decision to administer fibrinogen concentrate may benefit from close monitoring. Indeed, these situations may be analogous to other clinical scenarios where a decrease in fibrinogen is an early predictor of the severity of postpartum haemorrhage [33].

By adhering to institutional bleeding protocols, we ensured the close monitoring of patients throughout our study. Importantly, by focusing on evaluating the safety of fibrinogen concentrate administration in surgical settings rather than its efficacy, we could thoroughly examine adverse events, which is an often-overlooked aspect in other studies.

## 5. Conclusions

Our results indicate a favourable safety profile for fibrinogen concentrate in surgical patients, demonstrated by a low incidence of deaths and thromboembolic events, which is primarily attributed to other factors. However, future research should prioritize minimizing bias and enhancing statistical robustness to more effectively illuminate clinically significant patient safety measures.

## Figures and Tables

**Figure 1 jcm-13-06018-f001:**
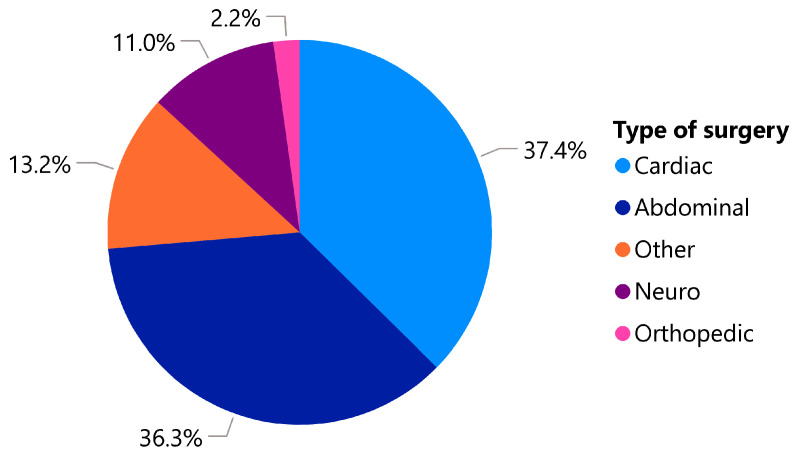
The proportions of patients by type of surgery.

**Figure 2 jcm-13-06018-f002:**
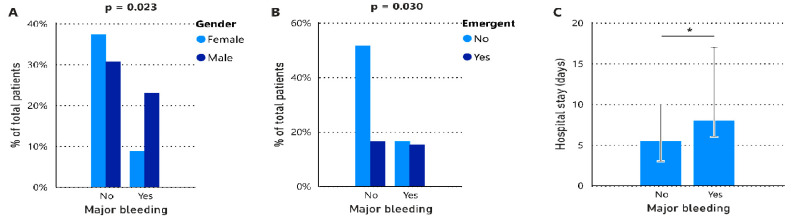
Distribution of patients by gender (**A**) and context of admission settings (**B**) according to the presence or absence of major bleeding. Bar plot showing with median with the IQR of the length of hospital stay for patients with and without major bleeding. Statistical analyses were performed using Fisher’s exact test (**A**,**B**) or the Mann–Whitney test (**C**). IQR, interquartile range (25th–75th). * *p*-value < 0.05.

**Figure 3 jcm-13-06018-f003:**
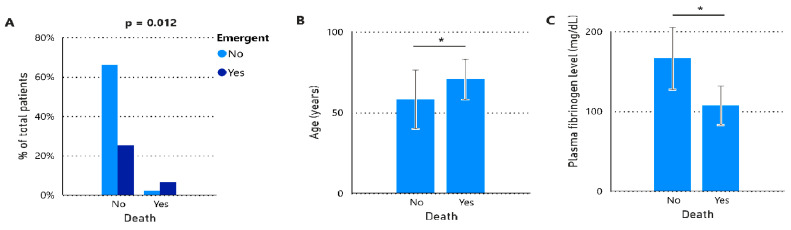
Distribution of patients by context of admission setting and death (**A**). Bar plots with mean ± SD of age (**B**) and plasma fibrinogen levels (**C**) according to the outcome of death. Statistical analyses were performed using Fisher’s exact test (**A**), Unpaired *t*-test with Welch’s correction (**B**), or unpaired *t*-test (**C**). SD, standard deviation. * *p*-value < 0.05.

**Figure 4 jcm-13-06018-f004:**
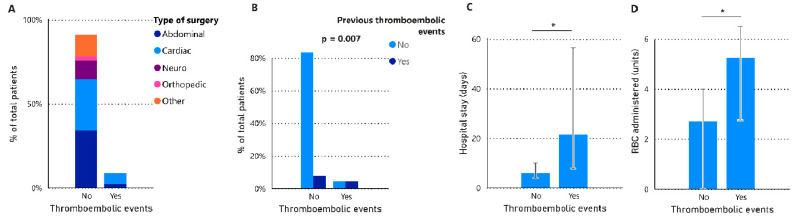
Distribution of patients by type of surgery (**A**) and history of previous thromboembolic events (**B**) according to the presence or absence of thromboembolic events during follow-up. Bar plots with median + IQR of the length of hospital stay (**C**) and units of RBC administered (**D**) in patients with and without thromboembolic events. Statistical analyses were performed using Fisher’s exact test (**B**) or the Mann–Whitney test (**C**,**D**). IQR, interquartile range (25th–75th). * *p*-value < 0.05.

**Figure 5 jcm-13-06018-f005:**
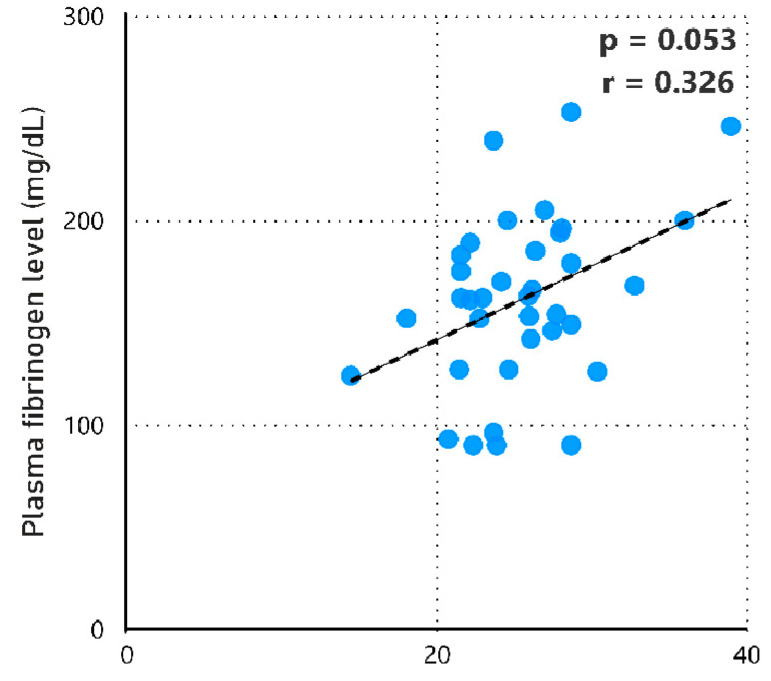
Spearman’s correlation of BMI with plasma fibrinogen levels (mg/dL) in all patients. The correlation coefficient, *p*-value, and best-fit line (dashed line) are displayed. BMI, body mass index.

**Table 1 jcm-13-06018-t001:** Demographic and clinical characteristics of all recruited patients.

	All Patients (n = 91)
Age, mean (SD), years	59.2 (18.2)
Males, n (%)	49 (53.8%)
Weight, mean (SD), kg	71.6 (15.3)
BMI, mean (SD), kg/m^2^	25.1 (4.2)
Previous thromboembolic events, n (%)	11 (12.1%)
Family history of thromboembolic events, n (%)	3 (3.3%)
Smoking, n (%)	20 (22.0%)
Cancer, n (%)	15 (16.5%)
Autoimmune disease, n (%)	15 (16.5%)
Other comorbidities, n (%)	35 (61%)
Oral contraceptives, n (%)	9 (9.9%)
Hormone replacement therapy, n (%)	1 (1.1%)
Anticoagulant therapy, n (%)	16 (17.6%)
Anti-aggregating therapy, n (%)	20 (22.0%)
Glucocorticoid therapy, n (%)	3 (3.3%)
Hgb levels pre-op, mean (SD), g/dL	12.9 (2.1) ^1^
Levels outside the normal range, n (%) ^a^	48 (55.8%) ^1^
Lowest Hgb levels post-op, mean (SD), g/dL	7.9 (1.6) ^2^
Levels outside the normal range, n (%) ^a^	89 (100%) ^2^
Plt levels pre-op, median [min–max], ×10^9^/L	216 [69–833] ^3^
Levels outside the normal range, n (%) ^b^	18 (23.4%) ^3^
Lowest Plt levels post-op, median [min–max], ×10^9^/L	139 [12–809] ^4^
Levels outside the normal range, n (%) ^b^	57 (71.3%) ^4^
Plasma fibrinogen levels, mean (SD), mg/dL	161.0 (41.9) ^5^
Levels outside the normal range, n (%) ^c^	33 (91.7%) ^5^
ROTEM MCF, mean (SD), mm	10.1 (2.8) ^6^
Thromboembolic events, n (%) ^d^	8 (8.8%)
Myocardial infarction	1
Deep vein thrombosis	2
Pulmonary embolism	2
Other	4
Mortality, n (%)	8 (8.8%)
No. of patients with platelet pool administered n (%)	26 (28.6%)
Units of Platelet pool administered, median [min–max]	0 [0–7]
No. of patients with prothrombin concentrate, n (%)	13 (14.3%)
Units of RBC administered, median [min–max]	2 [0–13]
No. of patients with FFP administered n (%)	36 (39.6%)
Units of FFP administered, median [min–max]	0 [0–8]
Doses of FC administered, mean (SD) [min–max], g	2.7 (1.4) [1–7]
Emergent surgery, n (%)	29 (31.9%)
Major bleeding, n (%)	29 (31.9%)
Type of surgery, n (%)	
Abdominal	33 (36.3%)
Cardiac/vascular	34 (37.4%)
Neurologic	10 (11.0%)
Orthopaedic	2 (2.2%)
Other	12 (13.1%)
Days of hospital stay, median [min–max] ^7^	6 [1–80]

^a^ Hgb normal range: 13.7–17.2 g/L; ^b^ Platelets normal range: 170–430 × 10^9^/L; ^c^ Plasma fibrinogen normal range: 210–400 mg/dL; ^d^ Patients who experienced more than one event were counted only once in the total. ^1^ n = 86, ^2^ n = 89, ^3^ n = 77, ^4^ n = 80, ^5^ n = 36, ^6^ n = 24, ^7^ n = 83. SD, standard deviation; Hgb, haemoglobin; Plt, platelet; ROTEM, rotational thromboelastometry; MCF, maximum clot firmness; FFP, fresh frozen plasma; and FC, fibrinogen concentrate.

## Data Availability

The original contributions presented in the study are included in the article, further inquiries can be directed to the corresponding author.

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
