# Peer review of "Safety of Fibrinogen Concentrate for Correcting Perioperative Bleeding-Associated Hypofibrinogenemia in Adults: A Single-Center Experience"

_jcm, 2024, doi:10.3390/jcm13196018_

Round 1

Reviewer 1 Report

Comments and Suggestions for Authors

I truly enjoyed reading real-life experience of fibrinogen concentrate supplementation in patients undergoing surgery. I have only minor comments, but that could significantly improve the manuscript:

1) how many patients were previously known /followed for hypofibrinogenemia, liver cirrhosis or other conditions that may affect low fibrinogen levels. Also it is of interest how many patients had known prior cytopenias or hematologic malignancy. This information may reveal in how many patients could low fibrinogen levels or impairment in other hemostatic factors be anticipated before elective surgery. Could authors provide this overview and account some of the analyses on this point.

2) do authors have information which particular fibrinogen concentrate was used in the study?

3) were same patients who experienced thrombotic events concomitantly those with major bleeding as well, i.e. how much overlap between these two categories was present?

Reviewer 2 Report

Comments and Suggestions for Authors

In introduction authors stated that: The primary objective of this study was to assess the safety of fibrinogen concentrate when administered within therapeutic ranges during the perioperative period to correct bleeding-associated hypofibrinogenemia in adults.

In objectives: The primary outcomes of interest in this study were overall mortality and the incidence of thromboembolic events at discharge

Plese cerify objectives

Metolodology:

1.      How is major bleeding defined? Were scores used in the evaluation of the degree of bleeding?

2.      Having in mind thet this was prospective study authors should specify cut off values for RBC and platelet transfusion as well as fibrinogen concetrate and thrombin complex concentrate. Authors shpuld specify in witch situation repeated doses were apllied, too.

3.      How DVT and PE, as well AIM were diagnosed? Is there any screening for them. For how long patients were followed for thrombosis? Three mounts is reasonable long period of time.

4.      How aticoagulant and antiplatelet therapy were managed? What was indication to applied this type of treapy?

Results:

1.      Please add more detals on major bleeding

2.      How many of patients with bleeding were terated with fibrinogen concetrate? How many of them were terated with RBC, Plt transfuison and other factors.

3.      Please add the normal values for paremeters in Table 1, as well as ranges and % of patients with abnormal values.

4.      Study group were defined as: The study included all adult patients (≥ 109

18 years of age) who underwent scheduled or emergency surgery associated with bleed ing coagulopathy and required the administration of fibrinogen concentrate. In the results section: A total of 91 adult patients who underwent surgery were included in the study. Plese defined group. In results you stated that during the perioperative period, 29 major bleeding were developed

5.      How many patient died due to bleeding or in group with bleeding

6.      How many patients who were terated with fibrinogen concetarte developed PE/DVT
